# Physical Activity and Diet Shape the Immune System during Aging

**DOI:** 10.3390/nu12030622

**Published:** 2020-02-28

**Authors:** Christopher Weyh, Karsten Krüger, Barbara Strasser

**Affiliations:** 1Department of Exercise Physiology and Sports Therapy, Institute of Sports Science, University of Giessen, 35394 Giessen, Germany; Christopher.Weyh@sport.uni-giessen.de; 2Medical Faculty, Sigmund Freud Private University, A-1020 Vienna, Austria; Barbara.Strasser@med.sfu.ac.at

**Keywords:** aging, exercise, nutrition, immunosenescence, inflammaging, kynurenine pathway

## Abstract

With increasing age, the immune system undergoes a remodeling process, termed immunosenescence, which is accompanied by considerable shifts in leukocyte subpopulations and a decline in various immune cell functions. Clinically, immunosenescence is characterized by increased susceptibility to infections, a more frequent reactivation of latent viruses, decreased vaccine efficacy, and an increased prevalence of autoimmunity and cancer. Physiologically, the immune system has some adaptive strategies to cope with aging, while in some settings, maladaptive responses aggravate the speed of aging and morbidity. While a lack of physical activity, decreased muscle mass, and poor nutritional status facilitate immunosenescence and inflammaging, lifestyle factors such as exercise and dietary habits affect immune aging positively. This review will discuss the relevance and mechanisms of immunoprotection through physical activity and specific exercise interventions. In the second part, we will focus on the effect of dietary interventions through the supplementation of the essential amino acid tryptophan, *n*-3 polyunsaturated fatty acids, and probiotics (with a special focus on the kynurenine pathway).

## 1. Introduction

Both the proportion of older people and the length of life have been increasing steadily in western societies. These demographic changes will significantly affect the economy, healthcare, and individuals’ well-being. Since aging is the highest risk factor for the majority of chronic diseases, an increasing number of people are living longer with impaired health and with disabilities. It is foreseeable that this development will significantly overstrain healthcare and social costs. Accordingly, specific lifestyle interventions are needed that aim to improve the health and quality of life of the aging population [1].

## 2. Aging and the Immune System

With advanced age, changes in all organs and tissues have been described, including the immune system. Components of the immune system are also influenced by age-associated changes occurring in such physiological systems as the endocrine, nervous, digestive, cardiovascular, and musculoskeletal systems, and vice versa. These changes, which affect both innate and adaptive immunity, significantly alter the constitution of leukocyte subsets and their major functions and are accompanied by a shift to a persistent proinflammatory status [2]. Because of this strong association between age and inflammation, the term “inflammaging” was introduced. Inflammaging describes a chronic condition of elevated proinflammatory mediators, including interleukin-6 (IL-6), tumor necrosis factor alpha (TNF-α), and interleukin-1 beta (IL-1) [3]. 

Clinically, immunosenescence has fundamental effects on health status. Besides an increased susceptibility to infections and a more frequent reactivation of latent viruses, a decreased vaccine efficacy in the elderly has been documented. For example, the yearly influenza vaccine is only 40–60% efficacious in persons aged 65 or above [4]. Increases in autoimmunity and cancer seem to also be related to an aging immune system. Therefore, the facilitation of various internal, orthopedic, psychological, and neurodegenerative diseases through immunosenescence and “inflammaging” has been intensively discussed [5]. 

### 2.1. The Innate Immune System during Aging

All cellular components of the innate immune system exhibit profound changes during aging. Monocytes, which represent about 5–10% of blood leucocytes, are classified into three subtypes. Classical monocytes express the surface marker CD14++, but are negative for CD16. Nonclassical monocytes express CD14+ and CD16++ and are distinguishable from intermediate (CD14++CD16+) monocytes. During aging, the number of nonclassical CD14^+^CD16^+^ monocytes increases, implicating a shift to a senescent, proinflammatory phenotype with short telomeres. Similarly, a shift in macrophage phenotypes has been documented. With regard to their opposite activities, macrophages are classified into proinflammatory M1 macrophages and more immunoregulatory M2 macrophages. During aging, a shift into the M1 phenotype has been documented [6]. Both processes have been suggested as contributing to inflammaging due to increased basal cytokine production [5,7]. Specifically, the increasing M1/M2 imbalance might be an important driver of age-related diseases. Accordingly, a significant relationship has been described between macrophage polarization and proneness to developing atherosclerotic plaques [8]. In neutrophils and monocytes/macrophages, receptor expression and even intracellular signal transduction are altered. Functionally, these senescence processes lead to diminished pathogen recognition, defective activation, decreased phagocytosis, and abnormal chemotaxis. Natural killer (NK) cells are innate lymphoid cells that represent about 15% of peripheral blood lymphocytes. While several NK cell subsets can be distinguished, a major characteristic of these cells is the expression of CD56. While CD56bright cells are immature cells providing regulatory functions, mature CD56dimCD16+ NK cells produce high levels of IFN-γ [9]. In elderly subjects, a remodeling of subpopulations has been documented. The progressive decrease of CD56bright cells reduces the amount of cells with mainly immunoregulatory functions. In parallel, highly differentiated CD56dim cells accumulate. Since NK cytotoxicity and cytokine production depend strongly on a balance between activating and inhibitory signals, these processes result in dysregulated cytokine production and reduced per-cell cytotoxicity in aging subjects [10]. 

### 2.2. Adaptive Immunity during Aging

While the total number of T-cells remains constant throughout life, there are considerable changes in the percentages of their particular subpopulations. With regard to the two main cellular subtypes, the number of CD8+ cells increases, while the number of CD4+ cells decreases, leading to a decrease in the CD4+/CD8+ ratio. This phenomenon is part of the immune risk profile (IRP) and indicates a condition of immunosuppression in certain diseases [11]. Within both populations, aging is accompanied by a reduction in T-cells with a naïve phenotype (expressing surface receptors such as CD45RA and CD28), while more differentiated T-cells accumulate. Naïve T-cells represent a pool of antigen-inexperienced cells that ensure an adequate immune responses against newly encountered pathogens. The reduction of naïve T-cells reduces the T-cell receptor (TCR) repertoire, mainly as a result of thymic involution and the process of cellular differentiation, which is driven by chronic antigenic stimulation and inflammation. In particular, human cytomegalovirus (HCMV) infection accelerates changes in both naïve CD4+ T-cell composition and in the accumulation of HCMV-specific CD8+CD28−T-cells [12].

After the age of 65, a shift to senescence and an accumulation of highly differentiated CD28−T cells have been described. These phenomena occur more strongly in CD8+ cells, leading to defective antigen-induced proliferation [13]. Senescent T-cells undergo replicative senescence because they have shortened telomeres and a reduced proliferative capacity [14]. These cells are strong producers of proinflammatory cytokines, aggravating “inflammaging” [7]. 

### 2.3. Inflammaging

Inflammaging describes persistent, nonresolving, low-grade inflammation during aging that fails to move to an anti-inflammatory status or repair tissue injury. Instead, it is progressive and seems to be strongly related to cellular immunosenescence [15]. The evidence suggests that inflammaging status affects the speed of aging and morbidity. Chronically increased levels of IL-6 and TNF-𝛼 in the elderly are associated with disability and mortality. They are closely related to various diseases, such as type II diabetes, cardiovascular disease, neurodegenerative disease, and cancer [16,17]. The differential regulation of IL-10 and TNF-𝛼 may be essential in predicting the progression of inflammaging. However, the mechanisms of age-related inflammation and its causal relationship with certain diseases are complex and mostly unclear [17].

## 3. Effects of Exercise and Diet on Immunity

Throughout life, the aging immune system adapts and is restructured in older individuals. The evidence suggests that these adaptive strategies, such as peripheral homeostatic proliferation of T-cells after involution of the thymus, cope with unique challenges, leading to successful immune aging. However, adaptive strategies might also fail, resulting in a maladaptive response and facilitating immune aging, inflammation, and disease [18]. We suggest that lifestyle factors, such as physical activity and diet habits, significantly affect the process of immunosenescence and inflammaging. Accordingly, regular exercise training and specific nutrition strategies support successful immune aging and decrease the risk of maladaptive immune aging.

### 3.1. Effects of Exercise on Immunosenescence 

Many findings have proven that a physically active lifestyle can have positive effects on the aging immune system [19,20]. In particular, regular exercise training seems to affect the aging processes of the innate as well as the adaptive part of the immune system. Regarding cells of the innate immune system, cross-sectional studies comparing older people with low fitness levels to physically active participants have indicated a number of advantages. Regular exercise in old age appears to be associated with improved NK-cell functioning [21]. Similarly, neutrophil functioning seems to be positively affected, since more active healthy seniors have a better migration of neutrophils toward IL-8 [22]. Besides data from cross-sectional studies, intervention programs have also indicated that exercise affects hallmarks of innate immunity. Accordingly, in one study, the number of nonclassical CD14+/CD16+ monocytes was reduced after a 12-week combined moderate strength and endurance training program, suggesting a reduction in more proinflammatory and senescent monocyte subtypes [23]. In patients with rheumatoid arthritis, high-intensive interval training was followed by an improvement of oxidative burst and bacterial phagocytosis in neutrophils [24]. Overall, results have indicated that increasing habitual physical activity enhances innate immune functions, which is indicative of reduced infection risk and inflammatory potential.

While data on the effects of exercise on the aging innate immune system have been limited, there are more studies available on the effects of physical activity on the hallmarks of immunosenescence in the adaptive immune system, especially in terms of T-cells. In early cross-sectional studies in the elderly, highly trained women showed improved mitogen-induced T-cell proliferation compared to an untrained control [21]. Improved T-cell proliferation was also reported in another study of elderly runners who trained for an average of 17 years, which was associated with improved functioning of the adaptive immune system [25]. Another study of healthy adults evaluated immune cell subpopulations in physical nonelite cyclists who engaged in high levels of physical activity, but no competitive sports, during most of their adult life. These adults, aged 55–79 years, were compared to inactive controls of the same age and to inactive young people aged 20–36 years. The older participants showed only a few signs of immunosenescence, including reduced markers of decreased thymus output (similar to young adults). Systemic inflammation and Th17-cell polarization were also reduced, and changes in the frequency of naïve T-cells and regulatory B-cells were not seen in the active elderly participants (compared to the older inactive subjects). However, the proportion of senescent CD28-CD57+ T cells was the same in both the active and inactive elderly [26]. Spielmann et al. found that aerobic fitness affects the age-related accumulation of senescent T-cells in peripheral blood [24]. In 102 male participants between 18 and 61 years of age, it was found that subjects with above-average peak oxygen uptake (VO_2max_) values had fewer senescent CD28-CD57+, CD4+, and CD8+ T-cells and an increased number of naive CD8+ T-cells compared to those with lower VO_2max_ values. This difference remained even after adjusting for age, body mass index, and body fat percentage. Interestingly, the authors were able to show that the association between age and senescent T-cells no longer existed when the data were adjusted for VO_2max_. These findings suggest that aerobic fitness may have a strong impact on changes in T-cells with aging [27]. Due to these findings, Minuzzi et al. recruited 19 master athletes >40 years of age with 20 years of training experience and compared their immune systems to an inactive control [28]. A reduction in senescent central memory (CM) and effector memory (EM) CD8+ T-cells and senescent and CM CD4+ T-cells was found in the active participants. In both the CD4+ as well as CD8+ subpopulations, the proportion of senescent highly differentiated effector memory -like phenotype (EMRA) T-cells was reduced in master athletes [28]. They discussed whether exercise only prevents the accumulation of these cell types over a lifespan or whether these cells are deleted through mechanisms such as apoptosis [29]. Minuzzi et al. confirmed their second hypothesis due to their results: that exercise induces cell death in apoptosis-resistant senescent T-cells [28]. This assumption was based on the finding that acute bouts of intensive exercise primarily promote the apoptosis of T-cells with a senescent phenotype [30]. Thus, various data from cross-sectional studies have revealed that regular physical activity is able to reverse age-associated changes in lymphocyte subpopulations and to partially reduce the age-related decline of T-cell functions. 

Some findings from these cross-sectional data have been supported by controlled exercise interventions, while others have not been successfully replicated. A study of overweight postmenopausal women aged 50–75 years found no changes in T-cell proliferation after 12 months of aerobic exercise [31]. Even a 32-week endurance and strength training exercise intervention did not increase T-cell proliferation in elderly participants [32]. In contrast, the increasing effect of regular endurance training on the CD4+/CD8+ ratio was repeatedly shown in older adults [33]. Similarly, after three weeks of endurance training (prediabetic subjects), a proportional increase in naïve and CM T-cells was found, while the proportion of senescent CD8+ EMRA T-cells decreased at the same time [34]. In this context, various variables have to be discussed. In this regard, the type of exercise might affect the process of immunosenescence. Positive results have largely been observed for endurance training, while the effects of strength training have been lacking [35,36]. The initial state of health might be relevant, too. While most studies have examined sedentary but healthy elderly individuals, participants with a “healthy risk” or an IRP may show greater benefits. Accordingly, a study of postmenopausal women after successful breast cancer treatment indicated an increase in T-cell proliferation as a result of regular aerobic training [37]. The results from the prediabetic collective (described above) indicate that individuals with a lowered state of health or with diseases with an IRP could ameliorate the hallmarks of immunosenescence more effectively by participating in regular exercise. Further studies are needed to evaluate the type of exercise, the dose–response relationship, and the potential for physical activity to have immune restorative effects.

When considering the already known effects of physical activity on clinically relevant outcomes, there is strong evidence with regard to vaccination. The benefits of maintaining thymic output and naive T-cells—such as those that have been shown to reduce immune senescence—were demonstrated in a study of 65–85-year-old men who engaged in regular physical activity for an average of 25 years. These participants showed higher antibody responses to influenza vaccination compared to controls of the same age [38]. Similarly, regular activity for three times per week at a moderate intensity over 10 months resulted in a significant increase in the antibody titer of an influenza vaccine [39]. Another study revealed that older active women showed greater antibody production against Flu B 18 months after vaccination compared to inactive women of the same age [40]. Various other interactions between immune aging and exercise training have been discussed in the context of specific diseases, such as cancer. Many tumor patients exhibit an advanced IRP. Considering that an active lifestyle reduces the risk of developing cancer and is associated with positive treatment outcomes in patients, it is possible that some of these effects could be due to the positive impact of regular exercise on immunosenescence [41].

### 3.2. Effects of Exercise on Inflammaging

It is now widely accepted that physical activity has an anti-inflammatory effect [42,43] and affects metabolic health in old age (positively) [44,45]. In contrast, epidemiological studies have shown that physical inactivity is associated with systemic low-grade inflammation and that physical activity and fitness are associated with a reduced concentration of various inflammatory cytokines in sera [46]. In this regard, a number of potential mechanisms that make physical activity an efficacious tool against systemic inflammation have been discussed. Since visceral adipose tissue seems to be an important source of inflammation, increased energy demand due to an active lifestyle seems to be an important mediator of immune-regulating effects. Due to reduced adipocyte size and metabolic stress, lower numbers of immune cells invade adipose tissue. However, it has been shown that exercise training directly affects the conversion of M1 into M2 macrophages, followed by a reduced secretion of proinflammatory cytokines [42,47]. 

With regard to inflammaging, toll-like receptors (TLRs) have been gaining increasing attention, because the upregulation of these receptors is associated with physical inactivity, systemic inflammation, and the development of age-associated diseases [48]. TLRs are transmembrane type I glycoproteins, which are mostly expressed in cells of the innate immune system [49]. They initiate an immune response after recognizing various exogenous signals, such as damage-associated molecular patterns (DAMPs), lipopolysaccharides (LPSes), or endogenous ligands such as heat shock proteins (HSPs), which are involved in chronic inflammatory processes [50,51]. The activation of TLRs typically induces a proinflammatory reaction and the release of cytokines [52]. In this context, the protective effect of physical activity is assumed to be a reduced expression of TLRs [22] followed by reduced proinflammatory activation. In particular, a reduced expression of TLR2 and TLR4 has been found after both acute as well as regular exercise [48,49,53,54].

Another mechanism that mediates the immune-regulating potential of physical activity seems to emanate from skeletal muscle itself. Cyclic muscle contractions and increased muscular energy metabolism leads to the production of various cytokines-termed myokines-or peptides with anti-inflammatory potential, indicating the muscle is an endocrine organ [55]. IL-6 has been identified as one of the most effective myokines in immune regulation. Levels of IL-6 increase during and after exercise, and the increases are proportional to intensity and duration [56]. The systemic secretion of IL-6 has a hormone-like effect in muscles and in other target organs and stimulates the production of immune-regulatory mediators such as IL-10 and the IL-1 receptor antagonist [57] and the downregulation of TNF by monocytes and macrophages [58]. Lifelong training also appears to affect the basal levels of pro- and anti-inflammatory cytokines. Results from middle-aged master athletes demonstrated that IL-1ra, IL-1β, IL-4, and IL-8 levels were elevated compared to an inactive, younger-middle-aged control group [59]. Furthermore, exercise-induced IL-6 was shown to inhibit endotoxin-induced TNF-α [42]. Besides IL-6, various other exercise-induced myokines that may affect the aging immune system have been explored. For example, the hormone meteorin-like has been shown to induce adipose tissue browning, increase IL-4 levels, and promote the polarization of M2 macrophages [60]. IL-7 [61] and IL-15 [62] are myokines that might stimulate lymphocyte proliferation, and it has been suggested that IL-7 exerts protective effects on the thymus. Both factors were found to be increased in elderly subjects who engaged in lifelong physical activity compared to their inactive controls [26]. IL-15 seems to have further effects on immune homeostasis, which is caused by the induction of a better survival rate of naive T-cells. At the metabolic level, IL-15 reduces the accumulation of visceral and white adipose tissue by reducing the accumulation of fat in preadipocytes (reviewed by Reference [20]). The upregulation of IL-7, with decreased levels of IL-6, due to lifelong physical activity has been suggested to have an important impact on thymus outcomes, while IL-15 has a significant impact on the naïve T-cell pool as well as a preventive effect on the development of proinflammatory adipose tissue. Taken together, some myokines are important mediators of immune functions in old age (Figure 1). These mechanisms play important roles in the prevention of and therapy for chronic low-grade inflammation. This particularly supports a role for exercise training as a nonpharmacological therapy to improve inflammatory status in patients with systemic autoimmune myopathies [63]. Therefore, physical activity could serve as an effective strategy against the development of inflammaging associated with an increased risk of age- related diseases such as cardiovascular disease, type 2 diabetes, and autoimmune disease.

### 3.3. Dietary Effects on the Immune Response 

There has been much interest in how dietary strategies can improve immunity in older people, and a nutritional approach is particularly suitable for the aging population since it requires less care than does a medical approach: it can also contribute to a more active lifestyle, thereby supporting well-being and active aging. Therefore, it is necessary to discuss recent developments in this field, as the elderly are more likely to have poor nutritional status, which further impacts their already impaired immune functioning. In the present review, we focus on the essential amino acid tryptophan, *n*-3 polyunsaturated fatty acids (PUFAs), and probiotics, with a particular interest in the kynurenine pathway due to the close relationship between kynurenine metabolism and inflammatory responses.

#### 3.3.1. The Essential Amino Acid Tryptophan 

Tryptophan is an essential amino acid found in many protein-based foods, including eggs, fish, dairy products, legumes, and meat. Levels of plasma tryptophan are determined by a balance between dietary intake and its removal from plasma as part of its essential role in protein biosynthesis [64]. The availability of tryptophan can be a key element in both cognitive functioning and mood because of its function as a sole precursor for serotonin [65]. The concentration of tryptophan in the blood (as a ratio to other large neutral amino acids (LNAAs)) is a marker of tryptophan availability for serotonin synthesis. However, even more importantly, tryptophan metabolism plays a pivotal role in immune system regulation [66]. For this reason, metabolism in the kynurenine pathway is the dominant metabolic fate for this essential amino acid. More than 95% of free tryptophan is degraded through the kynurenine pathway [67]. Degradation along this pathway is catalyzed by two main enzymes, indoleamine-2,3-dioxygenase (IDO) or tryptophan-2,3-dioxygenase (TDO). Whereas TDO is largely hepatic and induced by corticosteroids, IDO is ubiquitous and inducible by inflammatory stimuli. In particular, the Th1-type cytokine interferon-gamma (IFN-γ) induces various biochemical pathways, such as tryptophan breakdown [68]. Previous studies have shown enhanced tryptophan breakdown rates in the elderly and in several clinical conditions associated with increased proinflammatory immune activation [69,70]. Thus, an activated immune system in older adults can be detected by increased kynurenine/tryptophan levels, reflecting IDO activity (Figure 2). Although they have mainly been studied in relation to the brain, kynurenine metabolites generated by this strategy can affect several body compartments, inducing local and systemic adaptations (reviewed by Reference [67]). Chronic low-grade inflammation can lead to an elevation of circulating kynurenine levels, which can impact the central nervous system when neurotoxic compounds accumulate and proinflammatory cascades interfere with neurotransmitter receptors that control cognition and mood [71]. As mentioned above, a physically active lifestyle is known to increase the body's anti-inflammatory capacity by inducing anti-inflammatory cytokines and reducing proinflammatory cytokines. Consequently, chronic exercise interventions may lead to reductions in IDO activity as a result of anti-inflammation. Moreover, according to the preclinical study of Agudelo and colleagues, exercise induces the expression of peroxisome proliferator-activated receptor gamma coactivator 1-alpha-1 (PGC-1α1), which increases the expression of kynurenine aminotransferases (KATs) in skeletal muscle [72]. The KAT enzymes act to degrade kynurenine into kynurenic acid, which, in contrast to the parent compound, does not cross the blood–brain barrier. Thus, exercise acts to limit the exposure of the central nervous system to excess kynurenine, which has potential beneficial implications for mood and cognition. Even more profound are the effects of kynurenine metabolites on the immune system. The activation of the tryptophan-degrading enzyme IDO is an integral defense mechanism in the cell-mediated immune response, in which IFN-γ is the main activating factor. In addition to its role in innate immunity, the kynurenine pathway also plays a role in immunosuppressive and anti-inflammatory activities mediated primarily by T-cells of the adaptive immune system [69]. Kynurenine induces regulatory T-cell (Treg) development, and some tryptophan metabolites, such as 3-hydroxyanthranilic acid and quinolinic acid, have been shown to trigger selective apoptosis of Th1, but not Th2, cells [73]. In this way, a negative feedback loop evolves to prevent overwhelming immune reactions, and an immunotolerance status can be achieved. Accumulating evidence implicates gut microbiota in the regulation of kynurenine pathway metabolism, through which gut microbiota can influence brain function and behavior at the level of the central nervous system as well as local gastrointestinal functioning. Thus, interference with the microbiome is likely to influence the gut–brain axis [74]. The mechanisms may include alterations in microbial composition and tryptophan metabolism, immune activation, vagus nerve signaling, and the production of specific microbial neuroactive metabolites [75].

It has become more and more evident that tryptophan availability in kynurenine pathway metabolism can be modulated by changes in diet composition and lifestyle, such as physical exercise and weight loss [76,77,78,79,80]. Nevertheless, little is known about the effect of dietary tryptophan intake on the activated immune system of older individuals. Recently, we investigated the effect of a combined exercise–protein intervention on the kynurenine/tryptophan ratio and on neopterin concentrations in older patients during hip fracture recovery [81]. Perioperative nutritional interventions may have a positive effect on the response of the immune system during the early rehabilitation period following injury, as the majority of patients in this group suffered from malnutrition, negatively impacting patient outcomes [82]. We found that older patients with hip fractures demonstrated higher degrees of immune activation compared to reference values for healthy elderly individuals; however, protein enrichment (aiming at the consumption of 1.2 g protein/kg body weight per day) did not alleviate the Th1-type immune response in older patients during hospitalization. The results of this study further indicated that the lower tryptophan levels in hip fracture patients could not be attributed to low dietary intake, which was well above the recommended dietary intake of 250–425 mg/d [83]: rather, immune activation and inflammation conditions may have played a role, as tryptophan levels were found to be associated with IDO activity and neopterin concentrations, as well as with the serum level of C-reactive proteins. These results were in accordance with earlier findings that reported that inflammation causes the upregulation of IDO activity and leads to increased tryptophan catabolism via the kynurenine pathway [68]. On the other hand, in inflammatory arthritis and related disorders, kynurenine protects against the development of disease, while the inhibition or deletion of IDO increases its severity [84]. Similarly, the results from a recent study in hemodialysis patients suggested that the low tryptophan values in this patient group could not be attributed to high IDO activity or an inflammatory state [85]. In addition, the authors also found no association between dietary tryptophan intake and plasma tryptophan, which was not associated with all-cause mortality, while plasma albumin seemed to be a more important determinant of survival due to dialysis. Because most individuals (even older individuals and patients) consume adequate amounts of tryptophan, the beneficial effects of a diet rich in tryptophan and antioxidants are unlikely to be related to the greater availability of tryptophan [86], as was previously thought (this was based only on in vitro data) [65]. Further research is required to explore the biological roles of tryptophan and related kynurenine metabolites in the diet. At this time, only a few studies have assessed tryptophan in both the diet and plasma. As the availability of tryptophan depends on free albumin-binding sites, it would be advantageous to explore fatty acid profiles for any correlation. Furthermore, understanding the degree to which dietary tryptophan is actually absorbed in the gut-and hence, the role of the intestinal microbiota in controlling tryptophan availability and kynurenine metabolism-will considerably increase our knowledge of environmental factors and host–microbiome interactions.

#### 3.3.2. *n*-3 Polyunsaturated Fatty Acids

The process of aging is accompanied by chronic immune activation [66], and it is assumed that inflammation contributes to the development of sarcopenia, as proinflammatory cytokines directly affect muscle catabolic and anabolic signaling pathways [87,88]. Recently, we noted higher concentrations of inflammatory markers in older individuals with lower levels of handgrip strength, suggesting that inflammation may be involved in the loss of muscle strength [89]. On the other hand, muscularity per se may affect immune activation responses, as we found that improvements in maximum handgrip strength with a combined exercise–protein intervention were related to decreases in levels of neopterin (a sensitive marker of immune activation) in older patients during hospitalization [81]. Still, little is known about whether omega-3 polyunsaturated fatty acids (PUFAs) might be an alternative therapeutic agent for sarcopenia due to their anti-inflammatory properties and their role in impacting immune functioning in the elderly. The dietary intake of PUFAs has been linked to decreased inflammation through their effects on leukocyte action, eicosanoid production, and T-cell proliferation [90]. In general, there are fewer changes in muscle mass and function due to *n*-3 PUFA therapy (dosages from 2.0 to 3.3 g/day over a 3–6-month time period) in healthy older adults than there are due to resistance training [91,92,93]; however, a combined exercise–*n*-3 PUFA intervention was recently shown to trigger local anti-inflammatory and growth responses, thereby favoring skeletal muscle hypertrophy [94]. After six months of progressive resistance training (twice per week, three sets per exercise, at 50–85 % of one-repetition maximum (1 RM)) combined with an *n*-3 PUFA-rich diet (aiming at achieving an *n*-6/*n*-3 ratio < 2), but not after resistance training alone, gene expression of the proinflammatory cytokine IL-1β was downregulated, and that of the regulator of cellular growth mechanistic Target of Rapamycin (mTOR) was upregulated, in the skeletal muscle of older women. 

Studies on associations between the intake of *n*-3 PUFAs and individual metabolites related to the kynurenine pathway are scarce. The Western Norway B-Vitamin Intervention Trial provided data on the relationship between the dietary intake of fish or *n*-3 PUFAs and plasma concentrations of kynurenines, neopterin, and the kynurenine/tryptophan ratio in patients with coronary artery disease [95]. Although the associations were weak, the inverse associations between fish or *n*-3 PUFAs and neopterin and the kynurenine/tryptophan ratio may suggest slightly lower immune activation with a higher *n*-3 PUFA intake, which happened in the majority of patients who consumed more than the current recommendation of 250 mg/day [96]. A recent study in 12 young adults and 12 older adults provided metabolomic insight into the biological effects of *n*-3 PUFA supplementation (3.9 g/day over a four-month period) on protein metabolism and the kynurenine pathway [97]. The authors observed significant increases in hydroxyproline following supplementation in older adults, which was in line with previously reported significant increases in muscle protein synthesis following *n*-3 PUFA supplementation in the same individuals [98]. Furthermore, *n*-3 PUFA supplementation significantly reduced circulating kynurenine levels in healthy older adults who exhibited modest elevations in kynurenine (compared to younger adults). 

#### 3.3.3. Gut Microbiota and Probiotics

Accumulating evidence suggests that gut microbiota are linked to health status during the aging process [99]. A disturbed gut microbiome in the elderly is associated with chronic inflammation resulting from impairment of the mucosal barrier, which in turn affects various metabolic organs, such as the liver and adipose tissue, thereby contributing to metabolic inflammation [100]. Moreover, changing barrier function and nutrient requirements (with age) may influence skeletal muscle composition and function [101]. It has been reported that diet and exercise can modify the composition and diversity of gut microbiota [102,103,104] and may thus provide a practical means for enhancing gut and systemic immune functioning. 

Strategies to combat metabolic inflammation associated with alterations in the gut microbiome may include time-restricted feeding [105], diets rich in fibers [106] and restricted in carbohydrates [107,108], and further *n*-3 PUFA supplementation [109]. In addition, kynurenic acid, a downstream metabolite of the kynurenine pathway that is activated by exercise training, has been shown to promote the expression of genes involved in lipid metabolism and anti-inflammatory immune responses in adipose tissue [110]. 

Probiotics are interesting immunonutrients, since they demonstrate immunomodulation effects on both local and systemic immunity. Probiotics have been found to modify the population of gut microflora and have been shown to increase some aspects of mucosal and systemic immunity in healthy humans, such as altered cytokine production, increased natural killer cell cytotoxic activity, increased secretory immunoglobulin A (IgA) levels, and enhanced resistance to infections [111]. On the other hand, probiotics exert important anti-inflammatory “tolerogenic” effects that may reduce the burden of infection to a nondamaging level [112]. 

Some studies have established that probiotic intake can improve rates of recovery from gastrointestinal disorders and enhance resistance to upper respiratory tract infections (URTIs) in the general population [113,114]. Furthermore, probiotic intake can help to reduce the incidence of infection in athletes, and there is now evidence from a number of studies in support of this [115]. Probiotics could possibly have beneficial effects in preventing such issues in these individuals by modulating the gut microbiota, the mucosal immune system, and lung macrophage and T-lymphocyte functions [116]. Some of these effects appear to be connected to alterations in tryptophan metabolism. Recent findings have indicated reduced exercise-induced tryptophan degradation rates in subjects using probiotic supplements (1 × 10^10^ colony-forming units (CFUs) of multispecies probiotics, once per day for 12 weeks) [78]. This difference could have been due to the effect of probiotics on the microbiome composition in the gut, which may affect downstream immunoregulatory pathways [117]. Although this effect was not significant, daily probiotic intake was able to reduce URTI incidence. At the end of the study, the kynurenine/tryptophan ratio was significantly higher in individuals who developed infections compared to those who did not. The likely mechanisms of action for probiotics include direct interaction with gut microbiota, the stimulation of the mucosal immune system (both in the gut and at distant sites), and immune signaling between the different immune cells [118]. Although a daily dose of ~10^10^ CFUs (minimum of 10^6^ CFUs) appears to be the consensus, there is still some debate about the optimal duration of supplementation and the potential benefits of selecting and mixing specific bacterial strains (with or without probiotics). It has been proven in mice that the long-term consumption of fermented milk containing probiotic bacteria exerts immunomodulatory effects to maintain intestinal homeostasis without secondary effects [119]. In the elderly, the administration of fermented milk containing *Lactobacillus johnsonii La1* may contribute to suppressing infections by improving nutritional and immunological status [120,121,122]. Interestingly, in one study, while a high probiotic dose (5 × 10^9^ CFUs/day) resulted in significant increases in the percentages of activated T-suppressor (CD8+CD25+) and natural killer (CD56+ CD16+) cells, a low probiotic dose (5 × 10^8^ CFUs/day) increased activated T-helper lymphocytes (CD4+CD25+), B-lymphocytes (CD19+), and antigen-presenting cells (HLA-DR+) [123]. Thus, depending on the dose, probiotics may have different immune-enhancing effects in elderly subjects that might result in a better clinical outcome. 

## 4. Conclusions and Future Perspectives

Taken together, there is insufficient evidence to draw clear conclusions for specific dietary preferences based on gut and systemic immune functioning. Gut microbiota could represent a fundamental transducer of pro-anabolic stimuli from dietary intake to skeletal muscle [124]. On the other hand, exercise is an effective strategy for supporting successful immune aging and decreasing the risk of maladaptive immune aging. In this regard, regular endurance exercise seems to be the most promising approach to counteracting cellular immunosenescence and inflammaging. 

To guide clinical practice, future research should differentiate between short-term and long-term effects. Clinical trials need to establish the best frequency and duration of supplementation in order to discover any improvement in survival and disease-free periods, and further, to assess if specific bacterial strains are more suitable for prophylactic or therapeutic supplementation. This would enable a more personalized approach to treatment, which may help in enhancing the effectiveness and acceptability of currently available therapies. 

## Figures and Tables

**Figure 1 nutrients-12-00622-f001:**
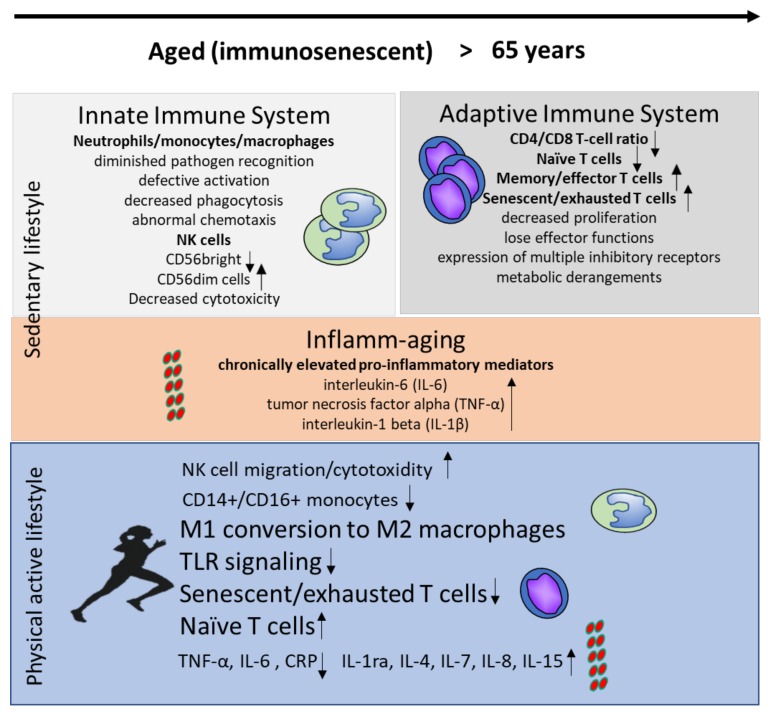
Immune changes in the innate and adaptive immune system during aging in inactive individuals and in response to a physically active lifestyle.

**Figure 2 nutrients-12-00622-f002:**
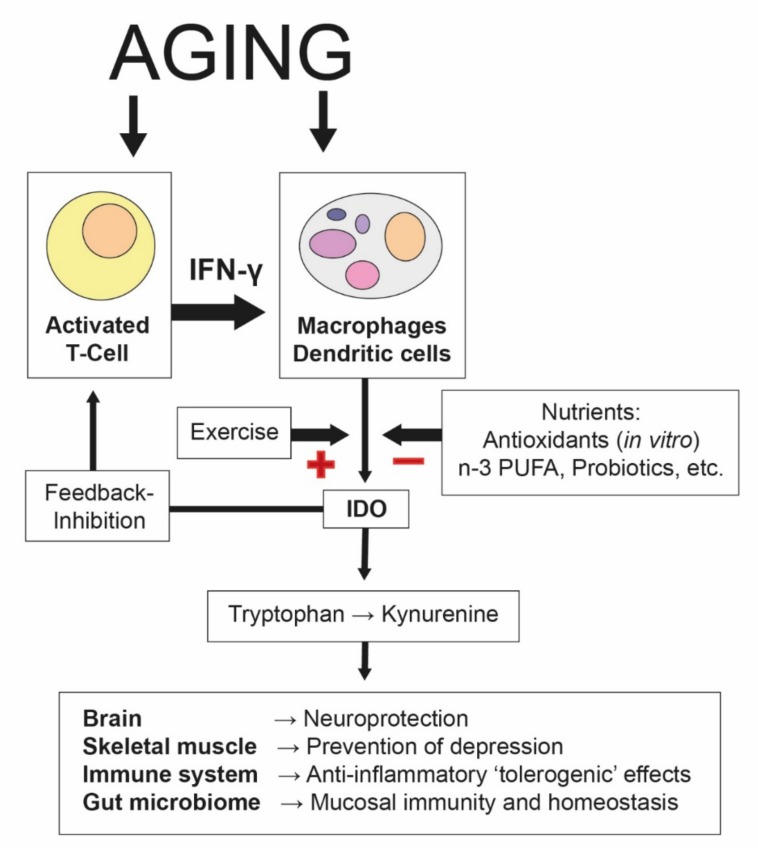
Aging-associated increases in indoleamine 2,3-dioxygenase (IDO). The process of aging involves proinflammatory pathways, in which the formation of Th1-type cytokine interferon-γ (IFN-γ) is of the utmost relevance. IFN-γ stimulates several enzymes, including indoleamine 2,3-dioxygenase-1 (IDO), which degrades tryptophan into kynurenine, an integral defense mechanism in the cell-mediated immune response. In addition to its role in innate immunity, the kynurenine pathway also plays a role in the regulation of the immune response by slowing down T-cell proliferation. Some tryptophan catabolites have been shown to be involved in the feedback inhibition of T-cell activation via regulatory T-cells, and thus immunosuppression. A supply of certain nutrients and bioactive compounds (such as antioxidants, *n*-3 PUFAs, or probiotics) can suppress (“−”) IDO activity and slow down Th1-type immune activation cascades. On the other hand, physical exercise stimulates the (“+”) induction of IDO, which is associated with an accelerated tryptophan breakdown and an increased kynurenine/tryptophan ratio [76]. Kynurenine metabolites can affect the brain and several other organs in peripheral tissues, where they induce local and systemic adaptations (see the main text for a description) [67]. Therefore, it is likely that kynurenine and its metabolites may contribute to the mediation of the health benefits of exercise and nutritional supplements.

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
