# Peer review of "Physical Activity and Diet Shape the Immune System during Aging"

_nutrients, 2020, doi:10.3390/nu12030622_

Round 1

Reviewer 1 Report

In the manuscript “Physical activity and Diet shape the Immune System during Aging”, the authors aim to provide information on the effects of exercise and diet on the immune function during aging. The authors generated a  comprehensive literature review consisting of 115 references which methodically provide information on the effects of aging on innate and adaptive immunity, and inflammation (“inflammaging”), then on the effects of exercise on immunosenescence and inflammaging, next on the influence of diets (tryptophan/kynurenine and n3 polyunsaturated fatty acids) on inflammation, and finally on the involvement of gut microbiota.

For the extensive scope of examined literature, the authors should be commended as the topic is of significant individual and public health. Just the accumulation of relevant archival literature is a significant contribution.

The weakness of the manuscript is its lack of basic integration and explanation of this rich source of information. Specifically:

Even readers with reasonable knowledge of immunology would benefit from an introductory section introducing the key players and functions of innate and adaptive immune systems. Providing an explanatory key of what CD4+, CD14+ (etc) do and which tribe of molecules they belong to would go a long way to provide a background against which the effects of immunosenescence , exercise and diets can be evaluated. There is a great vacuum between just listing in single sentences regarding what the individual studies uncovered, and an explanatory text that would tell the reader the physiological context of the findings. The reader really does not benefit from a “laundry list of references and brief statements about their results without being given an overall introduction On exercise (depending on its type) and the incidence of sequential pro-inflammatory and anti-inflammatory (protein building) effects. This would resolve some contradictory statements in the text to the effect that exercise is anti-inflammatory in one instance and pro-inflammatory in another instance. On what is known about inflammation-effects on tissues, effects of aging, effects of exercise or nutrients. There is a need for explanatory text so that the results of individual studies can be better understood. The section where the information is particularly confusing is the dietary section on tryptophan. We do not get any explanation on whether high tryptophan level is bad or good, whether its high level is useful simply to activate kynurenine pathway and increase the level of kynurenine, and the basic outcome of these changes. The two figures are just about useless. They seem to have marginal connection to the The second one does not explain the image symbols. Who is doing what? What effects are being produced? What are “+” and ” –“ symbols signifying? Why there a list 5 periphral tissues and systems in the rectangle below without telling us who is doing what to these? And in Figure 1, why is there an empty section on the left with the title “Young/immunocompetent” without any information for useful comparisons?

In summary, this review represents a lot of useful work in scanning the literature on the interactions of immune system with exercise and diet during aging. However, the review lacks explanatory and integrative elements which are in sore need of correction.

Author Response

Please see the attached letter for the authors responses.

Reviewer 2 Report

It is well written as is with references to realty recent paper on

the subject enclosed hereby.

Beyond medicine: Physical exercise should be always considered in patients with systemic

autoimmune myopathies.    

de Oliveira DS, de Souza JM, Shinjo SK.

Autoimmun Rev. 2019 Mar;18(3):315-316. doi: 10.1016/j.autrev.2018.11.003. Epub 2019 Jan 11. No abstract available.

Physical activity and autoimmune diseases: Get moving and manage the disease.

Sharif K, Watad A, Bragazzi NL, Lichtbroun M, Amital H, Shoenfeld Y.

Autoimmun Rev. 2018 Jan;17(1):53-72. doi: 10.1016/j.autrev.2017.11.010. Epub 2017 Nov 3. Review.

Interaction between food antigens and the immune system: Association with autoimmune disorders.   Vojdani A, Gushgari LR, Vojdani E.

Autoimmun Rev. 2020 Jan 7:102459. doi: 10.1016/j.autrev.2020.102459. [Epub ahead of print] Review.

The role of dietary sodium in autoimmune diseases: The salty truth.

Sharif K, Amital H, Shoenfeld Y.

Autoimmun Rev. 2018 Nov;17(11):1069-1073. doi: 10.1016/j.autrev.2018.05.007. Epub 2018 Sep 11. Review. Erratum in: Autoimmun Rev. 2019 Feb;18(2):214.

Author Response

For authors responses see attached document.

Round 2

Reviewer 1 Report

The revision is much improved in that, in most places, additional explanations help the reader.

1.There is however, still a massive confusion about the role of tryptophan and kynurenine which needs to be cleaned up it the readers are to get any benefit. Here is the problem as I see it: 

L 278-279: THE SECTION ONKYNURENINE CONTINUES TO BE CONFUSING. The following 2 statements appear contradictory and need more explanation:

L 278-279: :1”Previous studies have shown enhanced tryptophan breakdown rates in the elderly  and in several clinical conditions associated with increased immune activation [69,70] (what kind of immune activation? Pro-inflammatory?)

L 286-287: 2. “Tryptophan deprivation can suppress immune activation  processes via restriction of protein biosynthesis and the induction of regulatory T-cells by kynurenine  metabolites [72]” SO< WHICH IS  IT? REDUCED TRYPTOPHAN INCREASING IMMUNE ACTIVATION (STATEMENT 1) OR REDUCED TRYPTOPHAN SUPPRESSING IMMUNE ACTIVATION. THIS NEEDS TO BE STATED MORE CLEARLY. OTHERWISE, THE READERS ARE JUST BEING CONFUSED.

Figure 2 adds to confusion about kynurenine:L 3305-310:” On the other hand, physical exercise 306 stimulates (“+”) induction of IDO, which is associated with an accelerated tryptophan breakdown and 307 an increased kynurenine to tryptophan ratio [79]. Kynurenine metabolites can affect the brain and 308 several other organs in peripheral tissues where they induce local and systemic adaptations (see main 309 text for description) [67]. Therefore, it is likely that kynurenine and its metabolites may contribute in 310 the mediation of the health benefits of exercise and nutritional supplements” SO HOW DOES EXERCISE PRODUCE BENEFITS BY INCREASING KYNURENINE PRODUCTION?

Also,the feedback mechanism mentioned in Figure 2 is unclear and is not at all explained in the text.

2. Other confusing statements need explanation:

L347-348: WHAT DOES THE FOLLOWING MEAN?:” and sarcopenia may represent a consequence of a counter-regulatory strategy of the immune system” (WHAT IS BEING COUNTERREGULATED?)

L 355: WHAT IS NEOPTERIN DOING IN THIS REVIEW?:” decreases in neopterin levels”.

3. Figures are much improved but one needs additional fixing: 

Figure 1, top right rectangle: expression multiple inhibitory receptors> multiple inhibitory receptors expressed; central panel: chronic elevated pro-inflammatory > chronically elevated pro-inflammatory; “increased sign” needed in front of TNFalpha and IL-1beta;

4. There are MANY typographical/syntactic problems, as follows:

Abstract: Both the proportion of older people and the length of life increase(s) steadily

…..with impaired health and disabilities> with impaired health and with disabilities (not with impaired disabilities)

interventions are needed which aim to improve the health

section 2: shift into a persistent pro-inflammatory

However, a facilitation > Therefore, a facilitation

to their opposite activities macrophages> to their opposite activities, macrophages

of age related diseases> of age-related diseases

l 68: NK cells subsets>> cell subsets

l 72-73: In parallel 73 highly differentiated> In parallel, highly differentiated

l 73-74: Since NK cytotoxicity and cytokine production  depends strongly

l 109: immune-aging> immune aging

l 115: a physical active lifestyle> a physically active lifestyle

l 120: seems to be positive affected> seems to be positively affected

l 124: moderate strength> moderate-strength

l 160: This assumption based> This assumption is based

l 162: reverse age associated > reverse age-associated

l 166: others were not successful replicated> others were not successfully replicated

l 180-181: could ameliorate hallmarks of  immunosenescence more effective> more effectively

l 223-224: has been identified as  most effective myokines in immune regulation> has been identified as most effective myokines in immune regulation

l 225:¨ and  increase are proportional> and  increases are proportional

l 226: in muscle themselves> in muscle itself

l 236: it is suggested that IL 7 exert protective>  it is suggested that IL 7 exerts protective

L 259: the elderly is more likely> the elderly are more likely

l 246: a specific importance gains exercise training> a specific importance gains exercise training> particularly supports exercise training

L 277: interferon-gamma (IFN-   induces> Greek symbol for “gamma is missing as well as the en parenthesis.

327-328: Now it is stated that inflammation causes produc tion of kynurenine and not the other way around (kynurenine accumulation causing infklammation):” in accordance with earlier findings that have reported that inflammation causes upregulation of IDO 328 activity and lead to increased tryptophan catabolism via the kynurenine pathway [68].” PLEASE CLEAR THIS UP> IT IS A MESS.
